

# A systematic review of the validity of patient derived xenograft (PDX) models: the implications for translational research and personalised medicine

Anne T. Collins[1],[*] and Shona H. Lang[2],[*]

[1] Department of Biology, University of York, York, United Kingdom
[2] QED Biomedical, York, United Kingdom
[*] These authors contributed equally to this work.

## ABSTRACT

Patient-derived xenograft (PDX) models are increasingly being used in oncology drug development because they offer greater predictive value than traditional cell line models. Using novel tools to critique model validity and reliability we performed a systematic review to identify all original publications describing the derivation of PDX models of colon, prostate, breast and lung cancer. Validity was defined as the ability to recapitulate the disease of interest. The study protocol was registered with the Collaborative Approach to Meta-Analysis and Review of Animal Data from Experimental Studies (CAMARADES). Searches were performed in Embase, MEDLINE and Pubmed up to July 2017. A narrative data synthesis was performed. We identified 105 studies of model validations; 29 for breast, 29 for colon, 25 for lung, 23 for prostate and 4 for multiple tissues. 133 studies were excluded because they did not perform any validation experiments despite deriving a PDX. Only one study reported following the ARRIVE guidelines; developed to improve the standard of reporting for animal experimentation. Remarkably, half of all breast (52%) and prostate (50%) studies were judged to have high concern, in contrast to 16% of colon and 28% of lung studies. The validation criteria that most commonly failed (evidence to the contrary) were: tissue of origin not proven and histology of the xenograft not comparable to the parental tumour. Overall, most studies were categorized as unclear because one or more validation conditions were not reported, or researchers failed to provide data for a proportion of their models. For example, failure to demonstrate tissue of origin, response to standard of care agents and to exclude development of lymphoma. Validation tools have the potential to improve reproducibility, reduce waste in research and increase the success of translational studies.

Corresponding author
Anne T. Collins,
anne.collins@york.ac.uk

# INTRODUCTION

Advancing a candidate drug from preclinical testing into phase II clinical trials assumes that cancer models used in the laboratory are clinically predictive. Yet, over 90% of new drugs are ineffective in humans (*Johnson et al., 2001*; *Ellis & Fidler, 2010*) suggesting

that traditional preclinical models, such as cell lines cultivated in monolayer or xenografts derived from them, are a major factor in the low success rate of oncology drug development. A key consideration is the length of time these models have been in culture, undergoing extensive adaptation and selection and as such are unlikely to represent the heterogeneity and complexity of the disease.

In contrast, patient-derived xenograft (PDX) models, based on direct implantation of fresh cancer tissue specimens from individual patients into immunodeficient mice, are reported as more reliable models for preclinical research in many types of cancer (*Garber, 2009*; *Siolas & Hannon, 2013*). PDXs have been cited, in numerous studies, as better predictors of response; retaining cellular heterogeneity, architecture and molecular characteristic of the original cancer (*Garber, 2009*; *Tentler et al., 2012*). Nevertheless, there are challenges in using PDXs. For example, there are inconsistencies in take rates across tumour types, and importantly tumour grades, raising the question of whether PDXs are reflective of all cancer populations. Variability in take rate is also associated with mouse strain (*Ohbo et al., 1996*). The more immune-compromised strains appear to have more favourable take rates, but this is offset by the increased risk of lymphoma development (*Chen et al., 2012*; *John et al., 2012*; *Wetterauer et al., 2015*; *Taurozzi et al., 2017*), an under reported phenomenon in PDX research.

We sought to objectively assess the validity and reliability of PDX models as a platform for preclinical research in the four most common cancers: breast, prostate, colon and lung. Existing risk of bias tools do not interrogate how appropriate model selection is, nor how valid the models are. We previously developed novel tools to assess the validity of models, markers and the imprecision of results (*Collins, Ross & Lang, 2017*). This review concentrates on the assessment of the scientific quality of the studies, i.e., how well the models recapitulate the disease of interest, rather than the findings.

## METHODS

The methods for the literature searches and systematic review adhered to the Cochrane Collaboration guidance (*Higgins & Green, 2011*), to reduce the risk of bias and error. This study was reported according to the Preferred Reporting Items for Systematic Reviews and Meta-Analyses (PRISMA) statement (*Moher et al., 2009*), summarised in Table S1. The study protocol was registered with the Collaborative Approach to Meta-Analysis and Review of Animal Data from Experimental Studies (CAMARADES), http://www.dcn.ed.ac.uk/camarades/default.htm.

### Literature searches

Attempts were made to identify studies of PDX models of breast, colon, lung and prostate carcinoma. Searches in bibliographic databases were not limited by publication date, language or publication status (published or unpublished). Search strategies are presented in Table S2. The following databases were searched on 12 July 2017: Embase (OvidSP): 1974 –2017/07/11, Medline (OvidSP): 1946 –2017/06/WK5, Medline In-Process Citations & Daily Update (OvidSP): up to 2017/07/11, PubMed (NLM) (Internet) (http://www.ncbi.nlm.nih.gov/pubmed): up to 2017/07/03. The methods section of all

included articles and relevant reviews were also searched to identify studies for inclusion. The searches were performed by the authors.

## Inclusion and exclusion criteria

Inclusion and exclusion criteria are summarised in Table S2. We included original publications, which derived and validated PDX mouse models of human breast, colon, lung and prostate carcinoma. Specifically, we included the use of human tissue fragments or the use of primary human carcinoma cultures (≤3 passages) to generate xenografts in mice. Xenografts of any passage number were considered for inclusion. At least one validation assessment question had to be answered by the authors for inclusion; summarised in Table S3. We excluded xenografts generated from metastatic tissue, cell lines or those established in rats. Human cells, which had been genetically manipulated before xenograft generation, were excluded. PDX models that were purchased or validated elsewhere were excluded. Non-English language articles, conference proceedings, abstracts, commentaries and reviews were not included. Publications, which included primary and metastatic samples, were included and the primary samples alone were extracted where possible.

## Study selection, data extraction and data synthesis

Publications were loaded onto the systematic review web app, Rayyan, for title and abstract screening (*Ouzzani et al., 2016*). Titles and abstracts were independently screened by two reviewers. Articles meeting the inclusion criteria were obtained as full paper copies. Those were independently examined, in detail, by two reviewers to determine whether the full papers met the inclusion criteria of the review. All papers excluded at this second stage of the screening process were documented along with the reasons for exclusion. Any discrepancies between reviewers were resolved through consensus. Data extraction was performed by one reviewer and checked by a second reviewer. Any discrepancies were resolved through discussion. Studies were identified by the surname of the first author and by the publication year. Papers, which presented validation of the same PDX model, were grouped into 'studies'. A priori outcomes for extraction were primary outgrowth rate, established PDX rate and latency. During the course of the review, we also decided to investigate whether the PDX models could investigate tumour heterogeneity. A narrative summary of all the included studies was compiled.

## Quality assessment

Model validity was assessed by adapting the tool created by *Collins, Ross & Lang (2017)* (Table S3) and was defined as how well the PDX recapitulated the disease of interest. The number of PDX models (or established PDX lines) derived was compared to the number of models validated. We noted whether the authors had stated whether they followed the ARRIVE guidelines for the reporting of animal research (*Kilkenny et al., 2014*). Two reviewers independently assessed study quality and any discrepancies were resolved through discussion. The SYRCLE checklist was not employed as it is a risk of bias tool for interventional animal studies and was not appropriate (*Hooijmans et al., 2014*).

## RESULTS

### Literature searches and inclusion assessment

A summary of the identification and selection of studies for inclusion in this review is presented in Fig. 1, in accordance with the PRISMA statement (*Moher et al., 2009*). Literature searches of electronic databases retrieved 6,286 articles and hand searching identified 31 additional articles. After de-duplication 3,640 titles/abstracts were screened and 3,057 papers were excluded as having no relevance to the review. Full papers of 583 potentially relevant references were selected for further examination. Of these, 473 papers were excluded after reading the full paper; the reasons for exclusion are provided in Fig. 1 and a list of the excluded studies are provided in Table S4. 133 studies were excluded because, despite deriving a PDX, the authors did not perform any validation experiments. Lack of validation was defined as failure to provide evidence on: (a) tissue of origin, (b) confirmation that the PDX was derived from a given patient, (c) cell lineage, (d) confirmation that the PDX was derived from tumour and not normal cells, (e) absence of murine overgrowth, (f) comparable histopathology, (g) concordance for standard of care agents and (h) absence of lymphoma (Table S3). 110 records met the inclusion criteria; 29 for breast, 29 for colon, 25 for lung, 23 for prostate and 4 for multiple tissues. Some records provided validation methods for the same models, such records were grouped into 'studies'. The four records for multiple tissues provided additional information for each tissue. Overall, we identified 105 studies of model validations; 29 for breast, 31 for colon, 25 for lung, 20 for prostate. A list of the references for the included publications and the overall studies is provided in Table S5.

### Characteristics of PDX models

The characteristics of the PDX models are summarised in Table 1. The majority of studies reported on model development, whilst 10–42% reported the use of a PDX model to answer a biomedical research question (predominantly cancer research or drug discovery). A variety of mouse strains were used for derivation; the most common (>10% studies) were NSG, NOD/SCID, SCID and Balb/c nude. The engraftment site varied according to tissue. Breast models were predominantly orthotopic (55%) or subcutaneous (38%), whereas the majority of colon and lung models were derived from subcutaneous engraftment (94% and 75% respectively). The most common engraftment site for the derivation of prostate models was subcutaneous (55%) followed by subrenal (25%). Most models were derived from the engraftment of tissue fragments (69% to 87%) rather than isolated cells or primary cultures. In four studies reporting of methods was inadequate to ascertain whether tissue fragments or cells were used.

### Model validity

A model validity tool was previously created (*Collins, Ross & Lang, 2017*), and extended to include further judgements specifically for the PDX models (Table S3). Only one study (*Cottu et al., 2012*) reported that they had followed the ARRIVE guidelines (*Kilkenny et al., 2014*). Figure 2 summarises the overall judgements on the validity of the models. No study fully validated their reported models. This would require all signalling questions

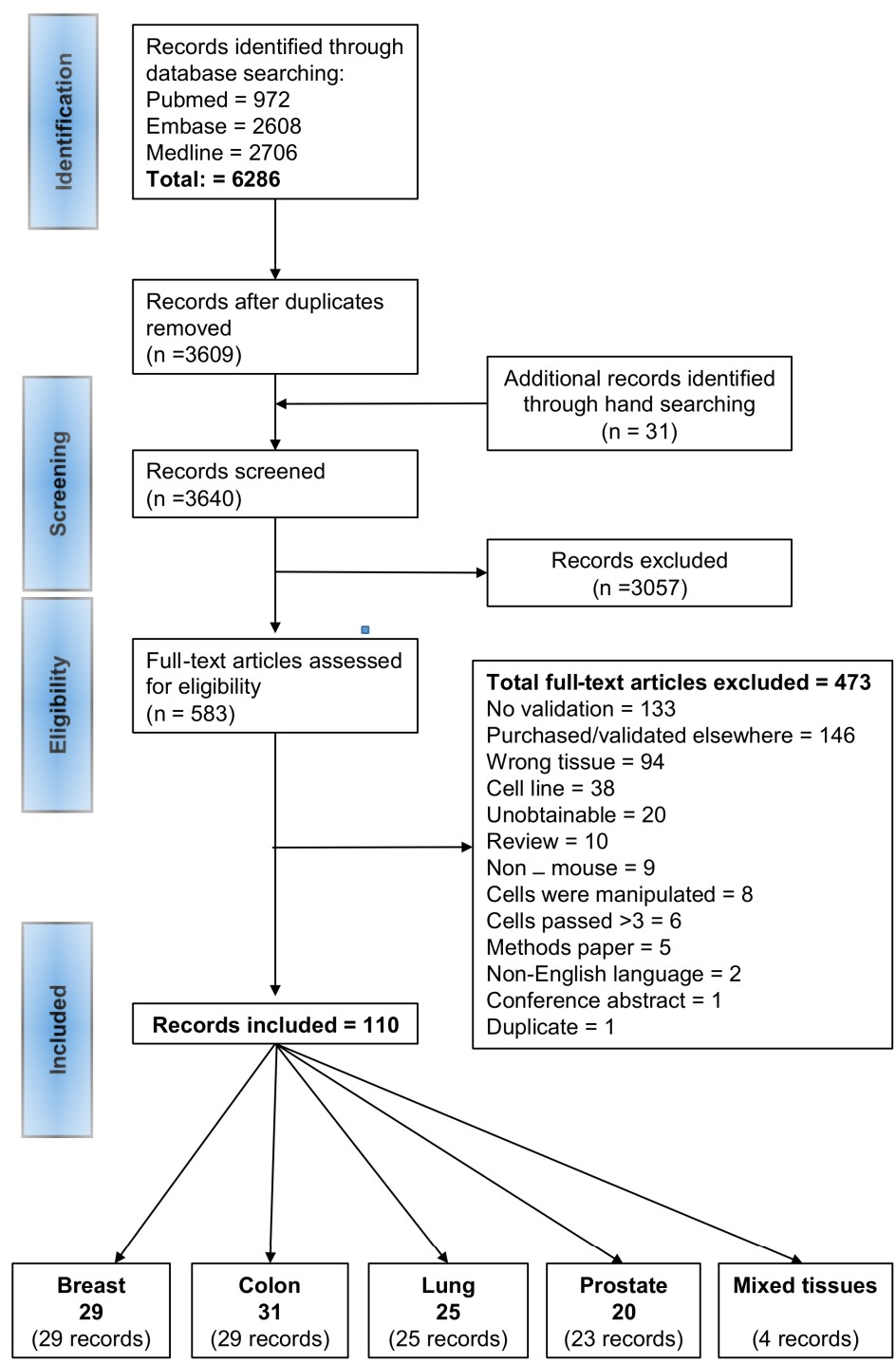

**Figure 1  PRISMA flow diagram of the study selection process.**

**Table 1** **Characteristics of PDX Models.** The number of studies reported is shown (percentage of total tissue studies).

| Tissue (Total studies) | Breast ($n = 29$) | Colorectal ($n = 31$) | Lung ($n = 25$) | Prostate ($n = 20$) |
|---|---|---|---|---|
| **Mouse Model:** No. studies (%) | | | | |
| B6D2F1 | 1 (3) | 1 (3) | 1 (4) | 0 |
| Balb/c nude | 4 (14) | 11 (35) | 4 (16) | 4 (20) |
| Balb/c nude; SCID | 1 (3) | 0 | 0 | 0 |
| Balb/c nude; NOD/SCID | 0 | 1 (3) | 0 | 1 (5) |
| SCID | 2 (7) | 2 (7) | 2 (8) | 6 (30) |
| SCID; NOD/SCID | 1 (3) | 0 | 0 | 0 |
| CD1 nude | 0 | 0 | 2 (8) | 0 |
| CD1 nude; SCID | 0 | 0 | 1 (4) | 0 |
| NCG | 0 | 0 | 1 (4) | 0 |
| NMRI nude | 1 (3) | 1 (3) | 0 | 1 (5) |
| NOD/SCID | 6 (21) | 7 (23) | 10 (40) | 3 (15) |
| NOD/SCID; NMRI nude | 0 | 0 | 1 (4) | 0 |
| NOD/SCID; NSG | 1 (3) | 1 (3) | 0 | 1 (5) |
| NOD/SCID; Rag2; NSG | 0 | 0 | 0 | 1 (5) |
| NOG | 0 | 1 (3) | 1 (4) | 0 |
| NSG | 9 (31) | 4 (13) | 2 (8) | 2 (10) |
| NSG; NOG | 0 | 0 | 0 | 1 (5) |
| NSG; NRG | 1 (3) | 0 | 0 | 0 |
| SCID/Bg; NGS | 1 (3) | 0 | 0 | 0 |
| Swiss nude | 0 | 2 (7) | 0 | 0 |
| **Engraftment of cells or tissue:** | | | | |
| isolated cells | 3 (10) | 2 (7) | 2 (8) | 0 |
| tissue fragments | 20 (69) | 27 (87) | 21 (84) | 17 (85) |
| isolated cells, tissue fragments | 2 (7) | 0 | 1 (4) | 3 (15) |
| primary culture | 1 (3) | 0 | 1 (4) | 0 |
| minced tissue | 0 | 1 (3) | 0 | 0 |
| unclear | 3 (10) | 1 (3) | 0 | 0 |
| **Engraftment site:** | | | | |
| orthotopic | 16 (55) | 0 | 2 (8) | 0 |
| subcutaneous | 11 (38) | 29 (94) | 18 (75) | 11 (55) |
| subcutaneous, subrenal | 0 | 1 (3) | 0 | 1 (5) |
| subrenal | 1 (3) | 1 (3) | 5 (17) | 5 (25) |
| subrenal, subcutaneous, orthotopic | 1 (3) | 0 | 0 | 3 (15) |
| **Use of PDX:** | | | | |
| Model establishment | 19 (66) | 19 (61) | 16 (58) | 18 (90) |
| Biomarkers/ cancer research/ drug discovery | 10 (34) | 12 (39) | 10 (42) | 2 (10) |

to be judged at low risk of concern. In breast and prostate, approximately half of the studies were judged to have high concern for model validity with the remainder judged as unclear. In contrast, only 16% and 28% of colon and lung studies, respectively, had high concern. Overall most studies were rated as unclear; this judgment was based on

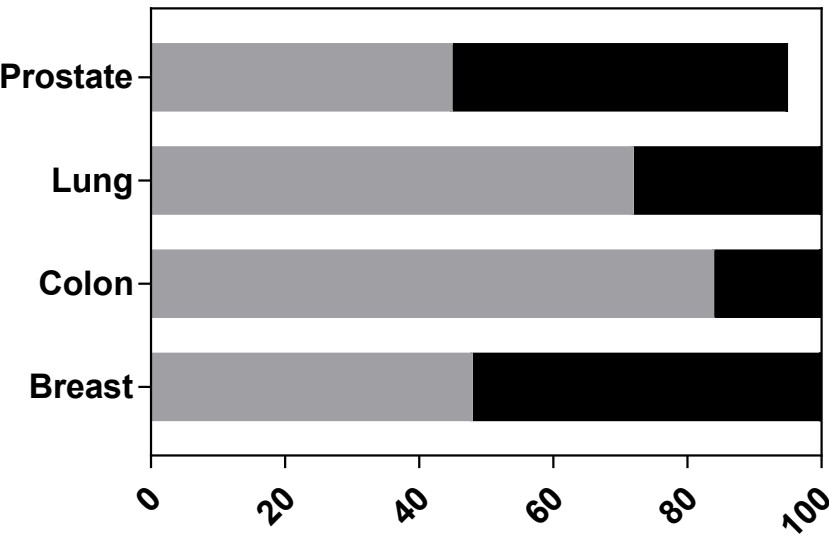

**Figure 2** **Overall validity ratings of PDX models.** Each study was assessed to determine if the reported PDX models were at high risk of concern for model validity. The graph indicates the percentage of studies per tissue: which had no concerns (white bars), high levels of concern (black bars), unclear levels (grey bars). 5% of prostate studies were not validated because they failed to derive a PDX.

a lack of information on one or more of the validation questions, but without high risk concerns.

The first five signalling questions of the model validity tool concentrate on how well the authors report methodology and sourcing of materials (Fig. S1). Such information is necessary to enable others to replicate and verify findings. The majority of studies provided ethical statements for the use of animal and human tissue; only 4% to 14% of studies did not. Similarly, most studies reported on source and strain of mice; only 6% of breast and 10% of colon studies did not. A clear description of how the mice and the xenografts were routinely maintained was not provided for 24% to 48% of all studies (dependent on tissue). All studies provided a description of how the PDX models were derived.

The second set of signalling questions judge how well the authors validated their models (Fig. 3 shows judgments per tissue type and Fig. S2 shows overall judgements). Individual prostate study data is presented in Table S3, to allow the reader to understand the judgements. Overall, most answers to any question were judged to be unclear, indicating that the authors either did not investigate the question(s) or only provided data for a proportion of their models. Indeed, more than half of all studies did not confirm tissue of origin, the presence of lymphoma or concordance of the model with the donor sample, whilst just over half of all studies (57%) confirmed that the xenografts were derived from tumour and not normal cells (Fig. S2). Colon had the fewest studies that were judged high risk. In contrast, at least one breast study had a high risk of concern for each validation question (Fig. 3). Analysis of the individual signalling questions indicated that for most studies there was a concern that the tissue of origin was not proven. In effect, 48% of breast studies, 16% of lung and 35% of prostate were classed as high risk because the models

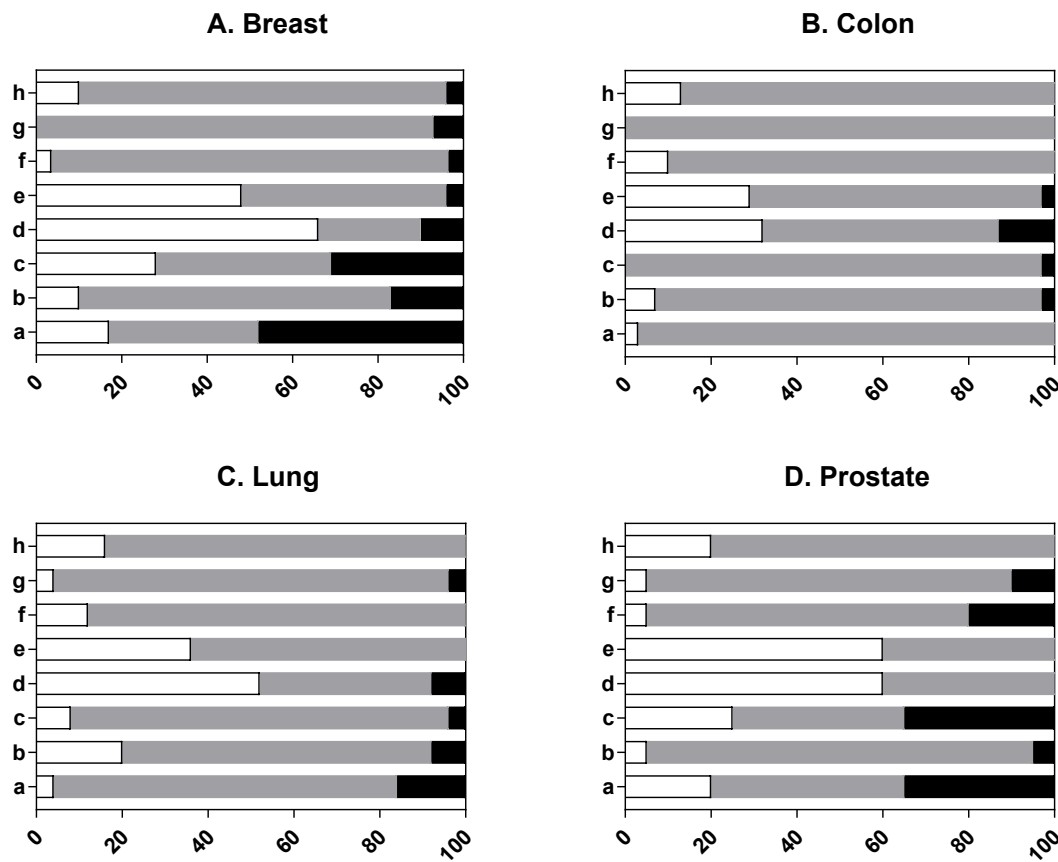

**Figure 3 Individual validity ratings of PDX models.** Each study was assessed to determine the level of concern for each signalling question eight criterion: a. was tissue of origin proven? b. confirmation that the PDX was derived from a given patient, c. was the cell lineage proven? d. confirmation that the PDX was derived from tumour and not normal cells, e. absence of murine overgrowth, f. was there comparable histopathology? g. concordance for standard of care agents? h. was the absence of lymphoma proven? A full description of signalling questions 8a–8h can also be found in Table S3 of the model validation tool. Each graph indicates the percentage of studies that were judged to be of low concern (white bars), high concern (black bars), unclear concern (grey bars). (A) Breast ($n = 29$ studies). (B) Colon ($n = 31$ studies). (C) Lung ($n = 25$ studies). (D) Prostate ($n = 20$ studies).

failed the authors own validation criteria for this question e.g., the xenografts did not express the stipulated tissue-specific markers. Likewise, most studies did not confirm that the PDX was derived from a given patient and 3 to 17% of all studies were judged as high risk because of the lack of concordant gene mutations or discordant clustering from gene expression studies. Most studies did not confirm that the PDX represented the cell type of interest (e.g., epithelial or neuroendocrine) and were classed as unclear, whilst 3 to 31% of studies had a high risk of concern as the PDX failed the authors own validation criteria. Most studies confirmed the tumorigenic nature of the PDX (32 to 66%) whereas a high risk of concern was found in 4% of breast studies, 13% colon and 8% lung. The majority of studies (29 to 60%) confirmed that human cells were present in the xenograft; murine overgrowth can occur with continuous passage (*Taurozzi et al., 2017*). In contrast, there

**Table 2  Frequency of Lymphoma Formation.**

| Tissue | References | Mouse model | Engraftment site | Sample origin | No. patients | % Biopsies forming Lymphomas |
|---|---|---|---|---|---|---|
| **Breast** | *Fujii et al. (2008)* | NSG | Subcut | primary | 57 | 2 |
| | *Bondarenko et al. (2015)* | NSG | Subcut | primary + mets | 3 | 33 |
| | *Wakasugi et al. (1995)* | Balb/c nude | Subcut | unclear | 5 | 80 |
| | *DeRose et al. (2011)* | NOD/SCID | Ortho | primary + mets | 42 | 2 |
| **Colon** | *Fujii et al. (2008)* | NSG | Subcut | primary | 48 | 38 |
| | *Bondarenko et al. (2015)* | NSG | Subcut | primary + mets | 7 | 28.5 |
| | *Mukohyama et al. (2016)* | NOD/SCID, NSG | Subcut | primary | 5, 8 | 20, 13 |
| | *Zhang et al. (2015)* | NOD/SCID | Subcut | primary | 43 | 2.3 |
| **Lung** | *Anderson et al. (2015)* | NOD/SCID | Ortho | primary | 10 | 10 |
| | *Ilie et al. (2015)* | CD1 nude, SCID | Subcut | primary + mets | 100 | 15 |
| | *Fujii et al. (2008)* | NSG | Subcut | primary | 2 | 50 |
| | *John et al. (2011)* | NOD/SCID | Subcut | primary | 157 | 12 |
| **Prostate** | *Fujii et al. (2008)* | NSG | Subcut | primary | 12 | 17 |
| | *Lin et al. (2014)* | NOD/SCID | Subrenal | primary; primary + mets | 16, 18 | 12.5, 11 |
| | *Wetterauer et al. (2015)* | NSG, NOG | Subcut; subrenal | primary | 27 | 80 |
| | *Klein et al. (1997)* | SCID | Subcut | primary | 3 | 33 |

**Notes.**
Subcut, subcutaneous; Ortho, orthotopic; mets, metastatic.

was a lack of confirmation that the histology of the donor tissue matched the corresponding PDX. This judgement was most often made because the authors failed to provide evidence for all the models or failed to report the methodology. The majority of studies did not validate whether the PDX replicated the patient response to standard of care treatment. This was largely due to a lack of reported data for this criterion, which is surprising given that PDXs are often reported as mimicking treatment response (*Garber, 2009*; *Tentler et al., 2012*; *Siolas & Hannon, 2013*). Moreover, 10% of prostate studies, 7% breast and 4% lung were considered high risk because of a lack of concordance with patient response.

Lastly, we assessed whether authors validated their models for the development of Epstein-Barr Virus (EBV)-associated lymphomas. We found that the majority (84%) of studies had not, therefore the risk was judged to be unclear overall. Only 15% of studies acknowledged if a PDX was not a carcinoma. Due to the heterogeneity between studies and the low number of studies reporting this occurrence we have summarised the results for each study and report the range of lymphoma development over the four tissue types (Table 2). For breast and prostate the frequency ranged from 2–80% whilst for colon the frequency ranged from 2–38% of biopsies. Although a range of mouse strains was used there was no indication that the rate of lymphoma was higher with the more immunocompromised strains. One study (*McAuliffe et al., 2015*) was judged high risk because the authors did not acknowledge a potential lymphoma and did not investigate further.

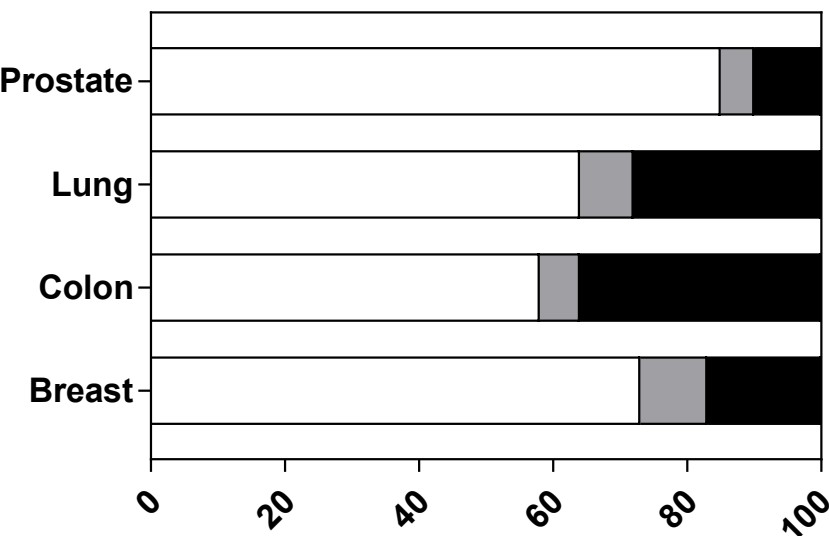

**Figure 4** **The proportion of PDX models from included studies that were validated.** Each study was assessed to determine if all reported PDX models were validated (any question 8 criterion). For each tissue type the percentage of studies is reported as: all models were validated (white bars), did not validate all reported models (black bars) or insufficient details reported to determine if all models were validated/not applicable (grey bars).

## Do authors validate all PDX Models?

We next assessed the proportion of PDX models from each study that were validated. This appraisal was based on whether the author attempted to answer one or more of the validation questions for all the models they derived. The results, summarised in Fig. 4, show that the majority of studies validated all reported models. Nevertheless, 36% of colon studies, 10% of prostate, 17% of breast and 28% of lung studies did not validate all published models. Examination of those studies with incomplete validation (listed in Table S6) indicate that 17 of 24 studies validated less than half of all derived PDX. For a small percentage of studies, it was unclear how many models were validated as the reporting was insufficient. One study was unable to derive a prostate PDX and this is reported as 'not applicable' (*Fujii et al., 2008*). These results indicate that users cannot assume that a published PDX model has been validated.

## Take rates of primary and established PDX models

We next considered the rates of primary and stable (e.g., PDX capable of serial transplantation) outgrowth from each tissue type. Take rate is either reported as the number of xenografts formed per patient or per sample. Reporting of take rate based on the number of samples (per patient) was the most difficult to assess because some studies engrafted multiple samples per mouse (most often found with subrenal engraftment) or multiple mice. If clearly reported by the authors we included both rates in Tables S7 and S8. Due to the heterogeneity between studies (e.g., method of engraftment, strain of mouse, donor pathology and definition of outcomes) we have summarised the range of take rates
**Table 3  Primary outgrowth rate and established PDX outgrowth rate.** Values based on primary tumour samples. Excluded are studies with high risk of model validity. Individual level study data are reported in Tables S7 and S8.

| Tissue | Range of reported Primary outgrowth rates (%) | Range of reported Established PDX rates (%) | Percentage of PDX forming stable lines (Median) |
|---|---|---|---|
| Breast | 10 to 31 | 5 to 27 | 55.8 |
| Colon | 14 to 100 | 10 to 41 | 68 |
| Lung | 26 to 90 | 9 to 83 | 71.1 |
| Prostate | 0 to 100 | 0 to 33 | 67 |

for each tissue type (Table 3). Excluded from this table are samples from metastases and studies at high risk for model validity.

The range of rates for primary outgrowths from breast cancer samples was 10 to 31.3%, based on four studies classed as unclear for risk of bias (a further seven reported this outcome but were judged to be high risk. The remaining breast studies were not included because the authors did not report this outcome). The rates reported for colon, lung and prostate xenografts were equally variable due to the heterogeneity between studies and lack of reporting for this outcome. Of the colon studies, 16 did not report primary outgrowth, only one was judged as high risk and the remaining 12 studies reported primary outgrowths between 14 and 100%. We considered five lung studies (17 did not report primary outgrowth and the remaining three were judged high risk) and seven prostate studies (7 were judged to be high risk and nine did not report primary outgrowth rates).

As indicated in Table S7 each study defined primary outgrowth differently, this included; exponential growth, any growth from implantation, tumours reaching a specific size, growth over a specified time (the latter was most often used with subrenal implantation). There was similar heterogeneity for the definition of stable growth which ranged from ability to passage at least once *in vivo* to >P5.

Some studies reported both primary and stable take rate, but this was not common; breast $n = 4$, colon $n = 5$, lung $n = 3$ and prostate $n = 3$. A comparison of the rates within studies showed that fewer stable lines were produced overall (Table 3).

Individual studies attempted to investigate the effects of changing the PDX methods. One lung study (*Russo et al., 2015*) found increased outgrowths from squamous carcinoma (92–96%) in comparison to adenocarcinoma (29–33%), different outgrowth rates were also reported according to hormone status and BRCA status of the donor (*Zhang et al., 2013*; *Risbridger et al., 2015*). Different outgrowth rates were reported between the engraftment of tissue fragments (50%) compared to isolated cells (12.5%; *Roife et al., 2017*). Such individual studies clearly indicate the importance of considering all these factors in the rate of outgrowth.

## Latency of primary outgrowths

Due to the heterogeneity between studies, we have summarised reported latencies from studies at low and unclear risk of bias for model validity (Table 4). The remaining studies either did not report latency or terminated the mice at a specific time irrespective of whether

**Table 4  Latency of primary outgrowths.**

| Study | Mouse strain | Engraftment site | Sample origin | Reported mean latency (days) | Reported median latency (days) | Reported range (days) | No. PDX |
|---|---|---|---|---|---|---|---|
| McAuliffe et al. (2015) | Balc/c nude | subcut | Breast | 109 | | 40–217 | 13 |
| Eirew et al. (2014) | NSG, NRG | ortho, subcut | Breast + Mets | | 217 | | 30 |
| Wakasugi et al. (1995) | Balb/c nude | subcut | unclear | | | 78–279 | 7 |
| Bogachek et al. (2015)[a] | Nude | subcut | Breast | 21 | | | 1 |
| Patsialou et al. (2012) | SCID | ortho | Breast | 140 | | 62–279 | 8 |
| Kabos et al. (2012) | NOD/SCID, NSG | ortho | Breast + Mets | | 119 | 73–228 | 10 |
| Davies et al. (1981) | Nude | subcut | Colon | 35 | | | 9 |
| Guan et al. (2016) | Balb/c nude | subcut | Colon | 94 | | | 340 |
| Jin et al. (2011) | Balb/c nude | subcut | Colon | 141 | | | 12 |
| Julien et al. (2012) | Swiss nude | subcut | Colon + Mets | | 59 | | 35 |
| Mukohyama et al. (2016) | NOD/SCID, NSG | subcut | Colon | 77, 76 | | 28–176, 19–223 | 3, 4 |
| Oh et al. (2015) | Balb/c nude | subcut | Colon + Mets | 90 | | | 150 |
| Mohamed Suhaimi et al. (2017) | NOD/SCID | subcut | Colon | | | 56–105 | 2 |
| Zhou et al. (2011) | Balb/c nude | subcut | NR | 21 | | 15–30 | 3 |
| Hao et al. (2015) | NOD/SCID | subcut | Lung | 112 | | 62–310 | 23 |
| Leong et al. (2014) | NSG | subcut | Lung | 104 | | | NR |
| Zhang et al. (2013) | SCID, nude | subcut | Lung | | | 20–95 | 14 |
| Roife et al. (2017) | NOD/SCID | subcut | Lung | 58 | | 26–175 | 9 |
| Lin et al. (2014) | NOD/SCID | subrenal | Prostate, +Mets | 570, 485 | | 93–1,147 | 9, 12 |
| Yoshikawa et al. (2016) | SCID | subcut | Prostate | 270 | | | 1 |
| Pretlow et al. (1993) | nude | subcut | Prostate | | | 60–330 | 10 |
| Klein et al. (1997) | SCID | subcut | Prostate | 300 | | | 2 |
| Terada et al. (2010) | nude | subcut | Prostate | 310 | | | 1 |

**Notes.**
[a]Derived from a primary culture.

Subcut, subcutaneous; Ortho, orthotopic; Mets, metastatic tumours; NR, not reported.

a tumour was palpable or not. Overall, colon (up to 223 days) reported the shortest latencies followed by breast (up to 279 days), lung (up to 310 days), whilst prostate had the longest latencies (up to 1,147 days).

## Tumour heterogeneity

We determined how many studies, developing PDX models from prostate cancer biopsies, had considered tumour heterogeneity by establishing multiple models from single patients, and whether all were validated. From the 20 prostate studies only 2 (10%) developed multiple PDX per patient and performed validation studies on all (Toivanen et al., 2011;

*Risbridger et al., 2015*). However, neither study produced stable lines from the multiple PDX. Five further studies developed multiple models per patient but did not validate all of them (*Wang et al., 2005*; *Priolo et al., 2010*; *Chen et al., 2013*; *Lin et al., 2014*; *Lawrence et al., 2015*). In summary, the majority of prostate studies did not derive multiple PDX per patient nor did they clearly report this information.

## DISCUSSION

One of the most serious obstacles confronting investigators involved in drug development is the failure of existing murine tumour models to reliably predict anticancer activity in the clinic. We assessed the evidence that PDX models more accurately reflect their human tumour counterpart by scrutinizing models based on a checklist of strict criteria. The major finding of this review was that half of all studies using prostate and breast PDX models were classed as high risk because they failed (evidence to the contrary) one or more of the validation questions. We found, mostly, discordance between a PDX and corresponding donor tissue for expression of tissue-specific markers, cell-specific markers and histopathology, demonstrating that some PDX models may not be what they claim to be. Overall, most studies were categorized as unclear because one or more validation conditions were not reported, or researchers failed to provide data for a proportion of their models. The most common reasons were; failure to demonstrate the tissue of origin, response to standard of care agents and exclusion of lymphoma.

This review highlights deficiencies in reporting. For example, 133 studies were excluded because the authors either failed to perform any validation experiments or simply did not report their findings. Whilst the majority of included studies validated all their models, to some extent, a number of published models were not validated. A lack of detail of methodology and vague reporting of results made it sometimes impossible to scrutinize evidence, all of which points to a failure of the peer review process. The ARRIVE guidelines, originally published in PLOS Biology (*Kilkenny et al., 2014*), were developed in consultation with the scientific community as part of an NC3Rs (National Centre for the Replacement Refinement & Reduction of Animals in Research) initiative to improve the standard of reporting of research using animals. Only one study stated that they followed the guidelines (*Cottu et al., 2012*), yet over 1,000 journals worldwide have endorsed them. It was not the aim of this review to closely assess whether authors presented data in accordance with the guidelines, but it is noteworthy that 14% of studies failed to provide an ethical statement and between 24–48% of studies did not provide a clear description of the routine maintenance of mice before or after xeno-transplantation.

The tool presented here provides an 'ideal set of validation criteria' for PDX models and can be adapted and applied to other models or marker studies. It may be unreasonable for a research group to provide evidence to fulfil all criteria, e.g., proving the tissue of origin of an undifferentiated (neuroendocrine) PDX is not straightforward. However, further studies should be undertaken to prove that the PDX matches the donor tumour. Similarly, studies that use primary outgrowth as their end-point (*Toivanen et al., 2011*; *Lawrence et al., 2015*; *Risbridger et al., 2015*), particularly for drug testing, must be able to demonstrate

targeting of malignant cells as normal cells can also populate grafts. Indeed, our assessment of primary and stable take rates show that up to 50% of primary outgrowths will not serially transplant.

Thirty-three percent of included studies reported the use of PDX models to investigate a research question in cancer research or drug discovery, highlighting the importance of rigorous validation of preclinical models. Through conducting this research, we aimed to highlight issues that may help alleviate the reproducibility crisis (*Baker, 2016*; *Ball, 2015*) and aid clinical translation. The use of systematic reviews highlights areas of weakness that can be improved going forward, but also provides a formal, unbiased and robust evaluation to provide guidance of the best evidence, or as in this review, the model that best fulfils a specific research need.

An important consideration for the use of PDX models in cancer research is tumour heterogeneity. Cancer, in an individual, is not a single disease. Tumours are heterogeneous that have evolved through a process of clonal expansion and genetic diversification, ultimately causing different prognoses within the same patient. The challenge for scientists and clinicians is to better understand this heterogeneity at a basic biological level and determine which subclones are of greatest risk to the patient (*Greaves & Maley, 2012*; *Bedard et al., 2013*; *Meacham & Morrison, 2013*). For PDX models it is therefore important to establish multiple models from each donor (with each clearly validated). Currently, the use of multiple models from individual patients is limited, particularly so in the prostate. More aggressive tumours are easier to propagate and are therefore over represented. It remains unclear why there is this selection bias for more aggressive tumours or indeed why prostate cancer is underrepresented. A consensus on methodology would help, but factors intrinsic to the sample are difficult to control for, such as uncertainty on the amount of viable tumour being engrafted. We were unable to perform a meta-analysis or funnel plot analysis of bias due to the high degree of heterogeneity between studies, but long latencies reported for prostate may be one of the reasons it is under-represented.

One of the major criticisms of PDX research is the lack of clear outcome definitions. Authors did not clearly define engraftment rate or experimental end-point. Some reported engraftment rate as the relative number of primary outgrowths or stable outgrowths (the latter defined as the ability to serially passage at least once). It was also unclear if rates were based on patient numbers or the number of samples. There was ambiguity on author's definition of 'successful' primary outgrowth. Success was based on tumour size or growth after a specific time interval. Such differences influence not only the reviewers' ability to synthesise the data but also the integrity of the data (*Brown et al., 2016*). Clearly, 'any growth' does not provide specificity particularly if that research has a clinical goal. PDX researchers should aim to have a set of agreed outcome definitions to improve the field; given that there is a EurOPDX consortium (http://europdx.eu/) it would be an important goal to implement.

We identified three other systematic reviews of PDX models; one followed PRISMA guidance (*Brown et al., 2016*) and two did not follow PRSIMA (*Jin et al., 2010*; *Lopez-Barcons, 2010*) *Brown et al. (2016)* reported on the large amount of heterogeneity in engraftment rates. None of the systematic reviews formally presented any quality assessment

of the primary studies; although SYRCLE was referenced by *Brown et al. (2016)*. All the reviews listed the validation techniques reported by the primary papers, but not whether these validation techniques were adequate or successful.

The search strategy had to be limited to the terms surrounding 'PDX'. However, it is likely that if it was expanded to including broader terms, such as 'explant' then we may have found additional relevant studies. Scoping searches indicated that broad terms would retrieve over 20,000 articles and was judged to be unfeasible. Whilst screening full papers for inclusion it was difficult to assess whether the same model had been used in different reports. Poor methodological reporting and a lack of a definitive name for the model prevented the reviewer from establishing multiple reports of a given PDX. This was felt to be especially the case if a PDX had been licenced to a company, who had likely re-named it and not reported its derivation. Similarly, we felt that the large number of studies which were excluded for not presenting any validation was, in part, due to poor reporting techniques and potentially the model had been validated but the results were just not reported.

## CONCLUSION

This is the first systematic review of PDX models to provide a comprehensive assessment of their validity using a novel tool (*Collins, Ross & Lang, 2017*), to assess quality based on empirical evidence. This is a major step forward as, until now, systematic reviews of biological models have provided a subjective assessment of key components of studies that the reviewers consider important (*Jin et al., 2010*; *Lopez-Barcons, 2010*; *Brown et al., 2016*) which does not allow a scrutiny of their worth. Existing tools critique the study design and are more appropriate to intervention studies (*Hooijmans et al., 2014*).

The use of systematic reviews to judge the reliability and validity of biomedical research will improve the success and reproducibility of subsequent translational clinical studies, particularly in this era of personalised medicine. Like similar evidence-based tools, this model validity checklist represents a dynamic document and is open to improvement. We invite others to comment on the tool and suggest improvements for the future.

### Funding
Shona H. Lang received no funding for this work. Anne T. Collins was supported by the Dutch Cancer Society Alpe d'HuZes/KWF program (UL2014-7058). The funders had no role in study design, data collection and analysis, decision to publish, or preparation of the manuscript.

### Grant Disclosures
The following grant information was disclosed by the authors:
Dutch Cancer Society Alpe d'HuZes/KWF program: UL2014-7058.

## Competing Interests

The authors declare there are no competing interests. Shona H. Lang is the co-founder of QED Biomedical, York, UK.

## Author Contributions

- Anne T. Collins and Shona H. Lang conceived and designed the experiments, performed the experiments, analyzed the data, contributed reagents/materials/analysis tools, prepared figures and/or tables, authored or reviewed drafts of the paper, approved the final draft.

## Data Availability

Raw data are provided in the Supplemental Files.

## Supplemental Information

Supplemental information for this article can be found online at http://dx.doi.org/10.7717/peerj.5981#supplemental-information.

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
