# Peer review of "A systematic review of the validity of patient derived xenograft (PDX) models: the implications for translational research and personalised medicine"

_PeerJ, doi:10.7717/peerj.5981_

## Round 0.1 · original submission · Minor Revisions

You will see from the two reviewers comments that they have suggested a number of minor changes and some more major changes, mainly by including a fuller explanation of the work.

I hope you take these as constructive comments and I urge you to address them fully in a revised manuscript.

Reviewer 1 ·

Basic reporting

a. The report is written in very clear English and is easy to read and well-explained
b. The literature cited is correct and at sufficient depth.
c. The article is well structured with appropriate use of Tables and Figures and all the raw data has been shared.
i. Modification 1: Could Figure 2 and 3 have different titles in the Figure legends to more easily discern the difference between them?
ii. Modification 2: Figure 4 does not appear to be described in the text of the results

Experimental design

a. The research question was well-defined and clearly stated
b. Use of appropriate standards was carried out or described in detail why they were not appropriate
c. The use of two independent assessors followed by resolving any discrepancies by consensus is appropriate
i. Modification 1: Figure 1 showing the PRISMA flow diagram has been uploaded incorrectly and appears to be from the authors previous paper (Collins, Ross and Lang 2017)
ii. Modification 2: The authors claim only one paper stated it followed the ARRIVE guidelines (results line 152 and discussion line 298). However, authors may well adhere to reporting their experiments in accordance with the ARRIVE guidelines without necessarily stating they are. Have the authors checked whether each paper actually did report their methods following the ARRIVE guidelines?

Validity of the findings

a. The data is robust and detailed. There are no methods employed to determine whether there are any statistically differences in level of concern between model type (breast, colon, lung and prostate) but this may not be appropriate in this setting.
b. The conclusions are well-stated and clearly explained.
c. The suggestions for improvement in validating and describing PDX models are detailed and give much food for thought researchers in the field of PDX generation

Additional comments

This paper aimed to systematically review generation of PDX models in breast, colon, lung and prostate cancer. They reported on the number of these papers that correctly assessed whether they had generated an appropriate PDX and crucially critiqued the models validity in a non-biased, systematic method. In general, this is an important paper for the PDX community to read due to its rigour and provides a salutary lesson in reporting and validation. I would recommend acceptance to publish with the modifications described above.

·

Basic reporting

In general, this paper is well written and structured

The numbers between the text and the flow-chart do not seem to match. For instance, the flow chart reports 34 included studies, while the texts suggests 110 (line 127)

Experimental design

No comment

Validity of the findings

Many studies were excluded after full-text reads. The reasons for exclusion are only shortly described. I encourage the authors to define clearly the exclusion reasons and give some practical examples (with literature reference) for some of the more common excluded types (e.g. “no asymmetric inheritance”, “no relevant organelle”, “wrong population”)

Additional comments

The authors should clarify what they mean with “model validation”. Apparently, publications without validation experiments were excluded for the review. I believe it is crucial that the authors describe what counts as “model validation” and what doesn’t.

Related to the comment above, I have no clue what the authors mean with “none of the studies clearly validated their models” (line 154-155).

It is unclear from this text what their newly created model validity tool (2017) is actually doing and how it was developed. I urge the authors to summarize that here in the current manuscript.

The authors report an important lack of information on one or more validation questions (line 159). It would be helpful if the authors could provide a frequency table with these questions and the times it couldn’t be answered.

---

## Round 0.2 · accepted · Accept

Many thanks for attending to the reviewers comments, which has improved the quality of your manuscript.